# Video Bokeh Rendering: Make Casual Videography Cinematic

RVR [34]                SteReFo [1]                DeepLens [28]

MPIB [17]               BokehMe [16]               Ours

**Figure 1: Given an all-in-focus video, we aim to render the refocusable video with a shallow depth of field, yielding aesthetically appealing results. Single-image-based methods [1, 16, 17, 28, 34] often suffer from flicker between frames and artifacts at edges. In contrast, our approach generates a temporally consistent video free from such artifacts. We encourage readers to experience the animations by viewing them with Adobe Acrobat or KDE Okular.**

## ABSTRACT

Bokeh is a wide-aperture optical effect that creates aesthetic blurring in photography. However, achieving this effect typically demands expensive professional equipment and expertise. To make such cinematic techniques more accessible, bokeh rendering aims to generate the desired bokeh effects from all-in-focus inputs captured by smartphones. Previous efforts in bokeh rendering primarily focus on static images. However, when extended to video inputs, these methods exhibit flicker and artifacts due to a lack of temporal consistency modeling. Meanwhile, they cannot utilize information like occluded objects from adjacent frames, which are necessary for bokeh rendering. Moreover, the difficulties of capturing all-in-focus and bokeh video pairs result in a shortage of data for training video bokeh models. To tackle these challenges, we propose the Video Bokeh Renderer (VBR), the model designed specifically for video bokeh rendering. VBR leverages implicit feature space alignment and aggregation to model temporal consistency and exploit complementary information from adjacent frames. On the data front, we introduce the first Synthetic Video Bokeh (SVB) dataset, synthesizing authentic bokeh effects using ray-tracing techniques. Furthermore, to improve the robustness of the model to inaccurate disparity maps, we employ a set of augmentation strategies to simulate corrupted disparity inputs during training. Experimental results on both synthetic and real-world data demonstrate the effectiveness of our method. Code and dataset will be released.

## CCS CONCEPTS

• **Computing methodologies** → *Computer graphics*; *Image manipulation*; **Computational photography**;

## KEYWORDS

Video bokeh rendering, Temporal consistency, Synthetic dataset

*ACM MM, 2024, Melbourne, Australia*
© 2024 Copyright held by the owner/author(s). Publication rights licensed to ACM.
ACM ISBN 978-x-xxxx-xxxx-x/YY/MM
https://doi.org/10.1145/nnnnnnn.nnnnnnn

## 1 INTRODUCTION

The bokeh effect is a popular photographic technique that creates aesthetic blur in out-of-focus areas, commonly used in movie shooting to highlight emphasized subjects. While DSLR cameras naturally produce a shallow depth of field effect, they are expensive, lack portability, and require expertise, making them less accessible for casual users. Smartphones, on the other hand, are widely used in our daily lives. However, limited by their small aperture sizes, they often produce videos with deep depth of field, lacking the aesthetically pleasing bokeh effects. Our work aims to bridge

the gap between smartphones and cameras, by rendering refocusable videos from all-in-focus videos, making casual videography cinematic.

Compared to video bokeh rendering, single-image-based bokeh rendering has been extensively studied, beginning with the classical rendering approaches [1, 19, 24, 27, 33, 34] that determine the blur radius from disparity map and focal point. However, disparity maps often contain errors, especially in regions with discontinuous depth, resulting in artifacts in the rendering results. To address this issue, neural rendering methods [16, 17, 28, 31] have been proposed to generate authentic bokeh rendering effects by learning from data.

Nevertheless, these methods are designed for single-image inputs, and applying them directly to each frame of the video often results in the following issues: (i) Flicker between frames arises due to the absence of spatial-temporal constraints; (ii) The inability to handle the disocclusion phenomenon (Fig. 2). The disocclusion phenomenon occurs when occluded objects in an all-in-focus image reappear during bokeh rendering with a large aperture. In a video sequence, the potential exists to recover information about occluded objects near the edge, which is necessary for bokeh rendering. Moreover, disparity maps for video sequences often contain flaws and temporal flickers, even when generated by state-of-the-art depth estimation methods. However, current single-image-based methods lack robustness to disparity maps, making them susceptible to producing artifacts. While a previous work, RVR [34], has proposed a video bokeh rendering system, they utilize optical flow to smooth the inputs of the disparity map. However, they still achieve bokeh rendering frame-by-frame using single-image-based models, thereby leaving inter-frame interactions and correlations untouched, as shown in Fig. 1.

To address the aforementioned issues, from a model design perspective, we present an end-to-end model for video bokeh rendering, termed Video Bokeh Renderer (VBR). The primary distinction between VBR and previous single-image-based bokeh rendering models lies in the capability to enforce temporal consistency and recover occluded information through a temporal fusion mechanism. To handle high resolution and large blur sizes, VBR adopts a coarse-to-fine framework comprising two sub-modules: a coarse bokeh generator and an iterative bokeh refiner. The coarse bokeh generator initially renders bokeh effects at a low resolution, and then the iterative bokeh refiner improves the result gradually. To harness information from adjacent frames for achieving aesthetically pleasing and temporally consistent video bokeh rendering results, we introduce the Temporal Fusion Block (TFB) into our model. Due to camera and object motions, identical spatial positions may not correspond to the same object across frames. Thus, in TFB, we first use deformable convolutions to align multiple features from adjacent frames and then utilize stacked convolution layers to merge the aligned features. Moreover, since disparity maps inevitably contain errors and flaws, we employ augmentation strategies during training to mimic inaccurate disparity maps. This enhances the robustness of our model to inaccuracies in disparity maps.

From the data perspective, due to the lack of paired all-in-focus and bokeh video data for training, we elaborate a video bokeh data synthesizing pipeline to generate the first video bokeh rendering dataset called Synthetic Video Bokeh (SVB). The training set of SVB comprises 3,000 videos, each containing 16 frames, resulting in a total of 144,000 frames. When generating video bokeh data, we mimic different shooting techniques, such as adjusting the focal plane, focusing on a moving target, or modifying the size of the aperture. Additionally, we synthesize a test set for validation purposes, comprising 300 videos and a total of 14,400 frames.

We evaluate our methods on both real-world data and the synthetic test set in the SVB dataset. The results demonstrate the advantages of the proposed video bokeh renderer. Our method achieves state-of-the-art results in rendering quality and temporal consistency on the synthetic test set. Moreover, our method shows robustness to inaccurate disparity maps and more stable performance compared with other methods. To further assess the quality of bokeh rendering from a subjective aspect, we also conduct a user study involving 55 participants on 20 videos collected from the internet. The comparison with other models reveals a preference among users for the rendering effects produced by our model.

In summary, our main contributions are as follows:

- We introduce VBR, the video bokeh rendering model that first leverages information from multiple frames to generate refocusable videos from all-in-focus videos.
- We propose the first synthetic video bokeh rendering dataset (SVB) using our ray-tracing-based video bokeh data synthesis method.
- Experimental results demonstrate that our method produces rendering results with better temporal consistency, enhanced edge rendering effects, and increased robustness to low-quality disparity maps.

## 2 RELATED WORK

### 2.1 Bokeh Rendering

Image-space-based methods [1, 16, 24, 28] operate on single RGB-D input in a post-processing manner and are typically divided into classical rendering methods [19, 27, 34] and neural rendering methods [16, 28, 31].

**Classical rendering methods.** Classical rendering methods [11, 19, 27, 34] determine the blur radius at different spatial positions based on the depth map and focal position. SteReFo [1] proposes a layer-based rendering method, dividing the input image into layers and applying a fixed convolution kernel to blur each layer. Although classical methods are efficient and can generate authentic bokeh effects in regions with smooth depth variations, they often produce artifacts in areas with discontinuous depth, such as edges.

**Neural rendering methods.** To address the artifacts at edges, neural rendering methods [16, 17, 28, 31] have been proposed to generate bokeh effects by learning from data. BokehMe [16] proposes a hybrid network to combine the classical renderer and neural renderer, leveraging the advantages of both techniques to address artifacts commonly encountered at edges in classical methods. To tackle the disocclusion phenomenon (Fig. 2), MPIB [17] employs an inpainting network to recover information of occluded objects in the all-in-focus image. However, relying on an inpainting network to fill in occluded information can introduce instability, potentially causing the model to collapse when the inpainting module fails.

While RVR [34] is the first work to introduce the notion of video bokeh rendering, it uses optical flow to smooth the disparity maps of input and renders the video frame by frame using a layer-based

classical method, overlooking the relationship between adjacent frames during rendering. In contrast, our approach addresses these challenges by leveraging information from neighboring frames to generate temporally consistent bokeh rendering results and mitigate artifacts at edges.

## 2.2 Temporal Fusion

Different from single-image bokeh rendering, achieving a satisfactory video bokeh rendering requires not only high rendering performance per frame but also temporal consistency. Moreover, bokeh rendering with a large aperture needs occluded information, further emphasizing the necessity of temporal fusion. Actually, leveraging and fusing information from adjacent frames has been a long-standing video understanding and processing [22, 25, 26]. 3D convolution [22] is used to learn spatial-temporal features. However, since adjacent frames are usually misaligned due to camera and object motions, directly fusing unaligned features may result in artifacts. Transformer-based temporal fusion [3, 23, 25] can bypass the feature alignment due to the global receptive field, but they often entail high computational costs. Another stream of work involves leveraging deformable convolutions to align frames before fusing them. Deformable convolutions [6] are widely employed for implicit feature alignment, as the learned offsets can overcome the fixed sampling position limitations in traditional convolutions and offer more flexible modeling capabilities. TDAN [26] introduces deformable convolutions into the network to align the reference frame and target frames at the feature level. Inspired by TDAN [26], subsequent works [4, 5, 29] start to align and fuse information from adjacent frames. Although they have shown promising ability in feature alignment and fusion, the heavy complexity and workload brought by the addition priors and complex structures remain undesirable. Additionally, inaccurate optical flow estimation can further compromise feature alignment. In this work, we also modulate cross-frame relationships with deformable convolution, but opt to discard the cascade structure and additional priors for a simpler and lighter-weight implementation.

## 3 PRELIMINARY

In this section, we illustrate some preliminary concepts related to video bokeh rendering. First, we analyze the disocclusion phenomenon employing a camera model. Then, we introduce the defocus map, which is used as input in our model to control the amount of blur at different spatial positions.

### 3.1 Disocclusion phenomenon

The disocclusion phenomenon refers that the occluded objects in an all-in-focus image can "re-appear" under a large aperture, *i.e.,* the occluded object can also contribute to the imaging process, indicating that the occluded information is necessary for rendering bokeh effects. As depicted in Fig. 2, the first row illustrates the imaging process of an all-in-focus image with a pinhole camera. As all the light must travel through the pinhole in a straight line, the blue object is blocked by the green foreground object. However, things are different when it comes to a large aperture, as shown in the second row. The rays from the occluded blue object can reach the green dot on the image plane because light can travel along other

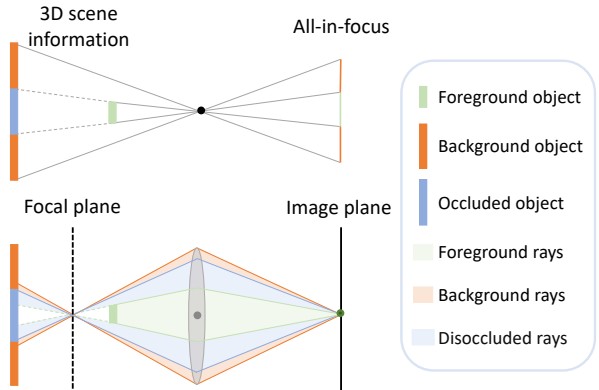

**Figure 2: Illustration of the disocclusion phenomenon. The first row depicts the imaging process of an all-in-focus image. Due to the occlusion by the foreground object (green), some information of the background object (blue) is lost. The second row of images illustrates the process of imaging with a large-aperture lens during bokeh rendering. Occluded areas become visible during light propagation, revealing information (disoccluded rays) that was previously lost in the all-in-focus image.**

routes within the large aperture lens. Hence, perfectly achieving bokeh rendering from a single image is an ill-posed problem, as the occluded information is necessary but absent, resulting in artifacts at edges. Fortunately, with a video sequence, there is potential to restore the occluded information from other frames. Therefore, our model utilizes the temporal fusion block to integrate information from multiple frames to deal with the disocclusion phenomenon, mitigating the artifacts at edges.

### 3.2 Defocus map

The defocus map provides the blur radius of each pixel, where any neighboring pixels within this radius contribute to the rendering result of the center pixel. The blur radius is determined by the depth and the focus point [27, 33]. Given the disparity of a pixel, its blur radius can be calculated by

$$r = K \left| d - d_f \right|, \tag{1}$$

where $d$ denotes the disparity of the pixel, $d_f$ represents the disparity of the focal position, $K$ indicates the degree of blur, and $r$ denotes the blur radius of the pixel. Each element in the defocus map corresponds to the size of the blur radius. The defocus maps $S$ can be calculated by the disparity maps $D$ and control parameters $K$ and $d_f$:

$$S = K \left( D - d_f \right). \tag{2}$$

## 4 METHOD

### 4.1 Overview

As illustrated in Fig. 3, the core of the video bokeh renderer is a cascaded model comprising: (i) a coarse bokeh generator that produces low-resolution coarse bokeh rendering results, (ii) an iterative

**Figure 3: Illustration of the Video Bokeh Renderer (VBR). VBR adopts a cascaded architecture to generate high-resolution results. First, a coarse bokeh generator renders bokeh effects in low resolution. Subsequently, the iterative bokeh refiner takes the all-in-focus frames, defocus maps, and the coarsely rendered results as inputs and gradually refines the coarse results into high-resolution outputs. To utilize information from multiple frames, we employ a temporal fusion block at the bottleneck of the sub-modules to fuse features from adjacent frames.**

refiner that progressively upscales and refines the coarse bokeh rendering results to high resolution. To leverage information from adjacent frames, we design a temporal fusion block positioned at the bottleneck of both the coarse bokeh generator and the iterative bokeh refiner. Our framework generates a bokeh video $B$ from an all-in-focus video $I$, a disparity video $D$, and controlling parameters. The controlling parameters comprise blur parameters $K$ and refocus disparities $d_f$, which respectively determine the overall amount of blur and the focal plane of each frame. Instead of directly inputting the disparity map and controlling parameters into the network, we utilize defocus maps to integrate the controlling parameters and disparity information, serving as inputs to the network.

## 4.2 Sub-modules for video rendering

**Coarse Bokeh Generator.** The coarse bokeh generator initially downsamples both the all-in-focus frames and defocus maps, simultaneously decreasing the numerical range of the defocus maps. This process yields rendering results with equivalent blur levels but at a lower resolution. The downsampling rate, denoted as $\alpha^{(0)}$, depends on maximum absolute value within the defocus map $S$ and the maximum blur capability $R_m$ of the module:

$$\alpha^{(0)} = min(1, \frac{R_m}{max(|S|)}) . \quad (3)$$

To preserve low-level details effectively, we integrate pixel unshuffle and pixel shuffle [20] within the sub-modules for downsampling and upsampling. Following the encoding of the input sequence into high-level features, the coarse bokeh generator utilizes a proposed temporal fusion block to integrate temporal information extracted from adjacent frames. Subsequently, the fused spatial-temporal features are passed into the decoder to obtain the coarsely rendered bokeh effects. We denote the output of the coarse bokeh generator as $B^{(0)}$.

**Iterative Bokeh Refiner.** To achieve aesthetic rendering results in high resolution, we design the iterative bokeh refiner to gradually enhance the rendered output. At the $t$-th iteration, the iterative

bokeh refiner improves and upscales coarse bokeh result $B^{(t-1)}$ to double the resolution. We also use the video sequence $I$ and the disparity maps $S$ as additional inputs to provide information. Before each iteration, we will resize the video sequence $I$ and the disparity maps $S$ to twice resolution of the coarse bokeh result $B^{(t-1)}$. The scale factor can be determined as follows:

$$\alpha^{(t)} = min(1, 2\alpha^{(t-1)}) . \quad (4)$$

In the first iteration, the coarse bokeh rendering results $B^{(0)}$ generated by the coarse bokeh generator are inputted into the iterative bokeh refiner. In each subsequent iteration, the bokeh rendering results $B^{(t-1)}$ from the iterative bokeh refiner become the inputs for the next iteration. The iteration ceases when the $\alpha^{(t)}$ in Eq. 4 equals 1, resulting in the final rendering result with the same resolution as the input.

## 4.3 Temporal Fusion Block

To enforce temporal consistency and mitigate disocclusion, we design a temporal fusion block to leverage information from adjacent frames. Due to the movement of the objects and the camera, the same spatial position in different frames may not correspond to the same object. Therefore, in the temporal fusion block, we employ deformable convolution [6] to align features from adjacent frames before applying stacked convolution layers to fuse them.

The detailed architecture is illustrated in Fig. 4. The temporal fusion block utilizes features extracted by the encoder as inputs. Given the features $f_{i-1}, f_i, f_{i+1}$, we concatenate the target frame and reference frames and pass them forward through convolutional layers to obtain offsets, which are then used to align the features:

$$\triangle O_{t+i} = g([f_{t+i}, f_t]), i \in \{-1, 1\} , \quad (5)$$

where $[\cdot]$ denotes the concatenation operation, $g$ denotes the convolutional layers responsible for predicting the offsets, and $\triangle O_{t+i}$ represents the predicted offsets used for feature alignment.

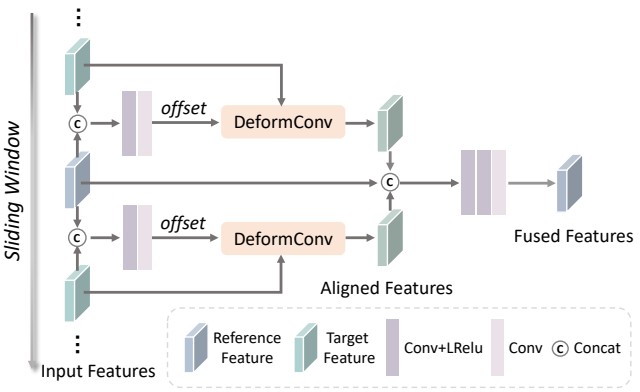

**Figure 4: Architecture of the Temporal Fusion Block. Deformable convolutions are employed to align features from adjacent frames, followed by fusion through stacked convolutional and activation layers.**

Then, the adjacent frames are aligned by deformable convolutions with the predicted offset:

$$\widetilde{f_{t+i}} = \text{DCN}([f_{t+i}, f_t], \triangle O_{t+i}), i \in \{-1, 1\} , \tag{6}$$

where DCN denotes the deformable convolutional layers utilized to align the features.

The temporal fusion block serves the purpose of aligning and merging information from adjacent frames, thereby reducing inter-frame flicker to maintain temporal consistency. Additionally, it enables the utilization of information from multiple frames to recover details in occluded regions, mitigating artifacts at edges caused by the disocclusion phenomenon.

## 4.4 Loss functions

We adopt the widely-used l1 loss and gradient loss as the spatial loss, where $n$ represents the position of the frame:

$$L_s(n) = L_1(B_n, B_n^*) + L_1(\nabla B_n, \nabla B_n^*) . \tag{7}$$

As for the temporal loss, we adopt the basic relation loss [7] to supervise temporal consistency:

$$L_r(n, n+1) = \left\| (B_{n+1} - B_n) - (B_{n+1}^* - B_n^*) \right\|_1 , \tag{8}$$

where $B^*$, $B$ indicate the ground-truth bokeh video and predicted video, respectively. The temporal constraint ensures temporal consistency by enforcing that the change values between consecutive frames in the rendering results match those in the ground truth.

The total loss is defined by Eq. 9, where $T$ represents the size of the sliding window,

$$L = \sum_{i=1}^{T} L_s(i) + \frac{\lambda}{T-1} \sum_{i=2}^{T} L_r(i, i-1), \tag{9}$$

where $\lambda$ is the weight factor and is empirically set to 5.

## 5 DATASET

Due to the challenges in capturing paired video data using DSLR cameras, currently, there is no dataset for video bokeh rendering. Existing datasets [8, 17, 34] for bokeh rendering only comprise image pairs and lack temporal information. To acquire paired bokeh videos for model training, we employ a synthetic method to generate pairs of all-in-focus and bokeh videos. In each video, we randomly select a background and some foreground objects, such as people or animals. Then, each foreground object will move along a random trajectory in the 3D world. Since the 3D coordinates of all the foreground objects and the background are known, following [17], we use the ray-tracing-based method to accurately produce bokeh effects in each frame, ensuring precise relation between adjacent frames.

**SVB Dataset Construction.** We utilize a landscape dataset [21] as the background and choose foreground objects from alpha matting datasets [9, 12–15, 18, 32]. To ensure data quality, we manually filter poorly annotated foreground objects and retain 1, 044 objects. The training set of SVB dataset consists of 3, 000 videos with accurate depth and bokeh rendering results, each containing 16 frames. In each video, 4 foreground objects are randomly chosen. To evaluate different models, we also synthesize a test set comprising 300 videos. To assess the ability of models to render complex scenes, we increase the number of foreground objects to 9, adding complexity to the test set. All the videos in the SVB dataset have a resolution of $256 \times 256$ pixels. In addition to the movement of the objects, we also mimic the process of changing camera parameters found in real-life scenarios, such as altering the focal plane and adjusting aperture sizes. Specifically, we achieve three types of bokeh techniques: (i) maintaining the focus target while varying the degree of blur, (ii) keeping the degree of blur constant while adjusting the focal plane, and (iii) maintaining the degree of blur while varying the focal plane from the farthest to the nearest or vice versa. These three control parameters encompass most bokeh effects observed in our daily lives. Please refer to the supplementary materials for detailed information and examples of the dataset.

## 6 EXPERIMENTS

### 6.1 Implementation Details

**Data Augmentation.** To enhance the robustness of our model, we introduce a set of augmentation strategies to simulate corrupted disparity maps during training. Specifically, we randomly apply Gaussian blur, dilation, erosion, and elastic transformation [2] to the disparity maps.

**Training.** We train the coarse bokeh generator and the iterative bokeh refiner sequentially. Each training sample is a video sequence with 4 consecutive frames, and the training batch size is set to 8. All the modules are trained for 10 epochs with Adam optimizer [10]. The learning rate is set to $10^{-4}$. We conduct all the experiments with Pytorch framework on two NVIDIA A6000 GPUs.

**Evaluation Metrics.** We evaluate both the video rendering quality and temporal consistency of different methods. Following [1, 16], we use PSNR and SSIM as the video rendering quality metrics. Moreover, we use $\text{PSNR}_{ob}$ and $\text{SSIM}_{ob}$ to evaluate the rendering quality at edges following [17]. For temporal consistency, we apply

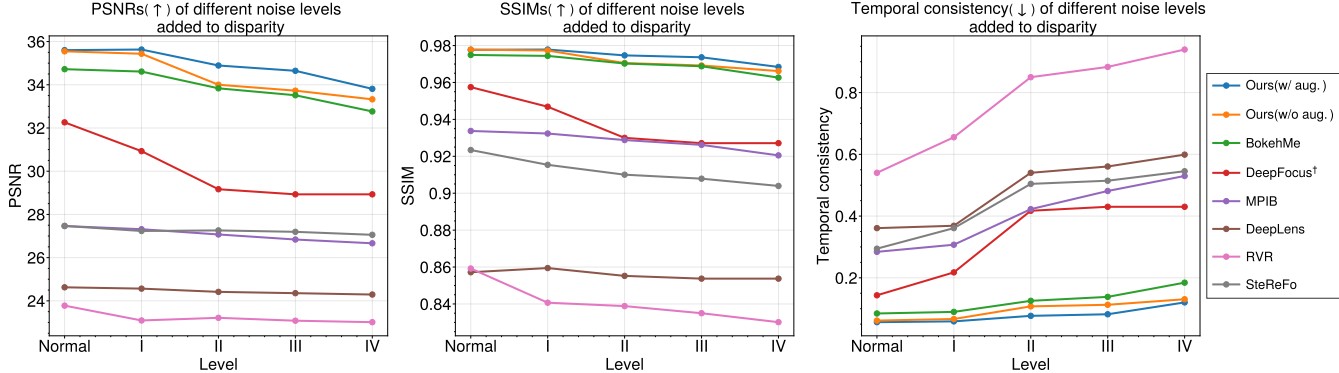

**Figure 5: Evaluation on the synthetic test set with corrupted disparity maps. Detailed information regarding the levels of disparity corruption is provided in Table. 2. Normal denotes that no noise is added.**

**Table 1: Quantitative results on the synthetic test set. The best performance is in boldface, while the second is underlined. Consistency denotes the temporal consistency defined in Sec. 6.1, here and after.**

| Methods | PSNR↑ | SSIM ↑ | $PSNR_{ob}$ ↑ | $SSIM_{ob}$ ↑ | Consistency↓ |
|---|---|---|---|---|---|
| RVR[34] | 23.8 | 0.859 | 22.3 | 0.878 | 0.540 |
| SteReFo[1] | 27.5 | 0.923 | 25.6 | 0.882 | 0.294 |
| DeepLens[28] | 24.6 | 0.857 | 27.0 | 0.944 | 0.361 |
| DeepFocus† | 32.3 | 0.958 | 28.2 | 0.939 | 0.143 |
| MPIB[17] | 27.5 | 0.934 | 23.6 | 0.875 | 0.284 |
| BokehMe[16] | 34.7 | 0.975 | 28.9 | 0.925 | 0.084 |
| Ours | **35.6** | **0.978** | **30.6** | **0.951** | **0.056** |

the basic relation loss defined in Eq. 8 on the whole video as:

$$\frac{1}{N-1}\sum_{i=2}^{N} L_r(i, i-1),\qquad(10)$$

where $N$ denotes the length of the video.

## 6.2 Results on Synthetic Video Bokeh Dataset

To verify the performance of our proposed method, we compare the effectiveness and the robustness of our model, VBR, with two classical bokeh rendering methods: RVR [34] and SteReFo [1], and four neural rendering methods: DeepFocus [31], DeepLens [28], MPIB [17] and BokehMe [16] on the proposed synthetic test set. We found that DeepFocus [31] sometimes collapses under parameter settings with large blur size. Therefore, we resize the inputs of DeepFocus [31] to a range that the network can handle, and then upscale it back to the original resolution for evaluation. This modified method is marked with a superscript †.

**Quantitative experiment.** As shown in Table. 1, our model achieves the highest PSNR and SSIM scores, indicating better rendering quality. Moreover, our method shows better rendering quality at edges compared with BokehMe, according to $PSNR_{ob}$ and $SSIM_{ob}$. These results further demonstrate that extracting occluded information

**Table 2: Levels of disparity corruption. The scale shift is designed to mimic the disparity's flicker on the overall scale. We randomly choose a scale factor within $[0.9, 1.1]$ and multiply the disparity with this scale factor.**

| Level | Gaussian blur | Erosion/Dilation | Elastic Transformation[2] | Scale Shift |
|---|---|---|---|---|
| I | ✓ | | | |
| II | ✓ | ✓ | | |
| III | ✓ | ✓ | ✓ | |
| IV | ✓ | ✓ | ✓ | ✓ |

from adjacent frames can help mitigate artifacts at edges, *e.g.*, addressing the disocclusion phenomenon proposed in Sec. 3.1. Our method also achieves the best temporal consistency, with a 33% relative improvement compared to BokehMe. This result further reveals the benefits of temporal fusion in reducing flicker between frames to generate temporal consistent results.

**Robustness.** To validate the model's robustness to inaccurate disparity maps, we test various methods with impaired disparity maps as inputs. Further details regarding the disparity corruption can be found in Table. 2. As illustrated in Fig. 5, VBR outperforms other methods in all metrics under different levels of corrupted disparity settings, indicating its superior robustness to disparity maps. Additionally, in Fig. 6, even in zones with inaccurate disparity, our method can still generate aesthetic bokeh effects.

## 6.3 User Study on Real-World Videos

Due to the subjective nature of perception towards video bokeh rendering results, we conduct a user study on real-world videos to better evaluate different methods from the subjective perspective. We collect 20 all-in-focus videos from the internet with a resolution of $1080 \times 1920$, featuring people, natural landscapes, and other diverse themes. Since there is no ground-truth of the disparity, we obtain the disparity maps by a depth prediction model [30].

**Quantitative results.** We use different methods to render the videos under the same controlling parameters. During the test,

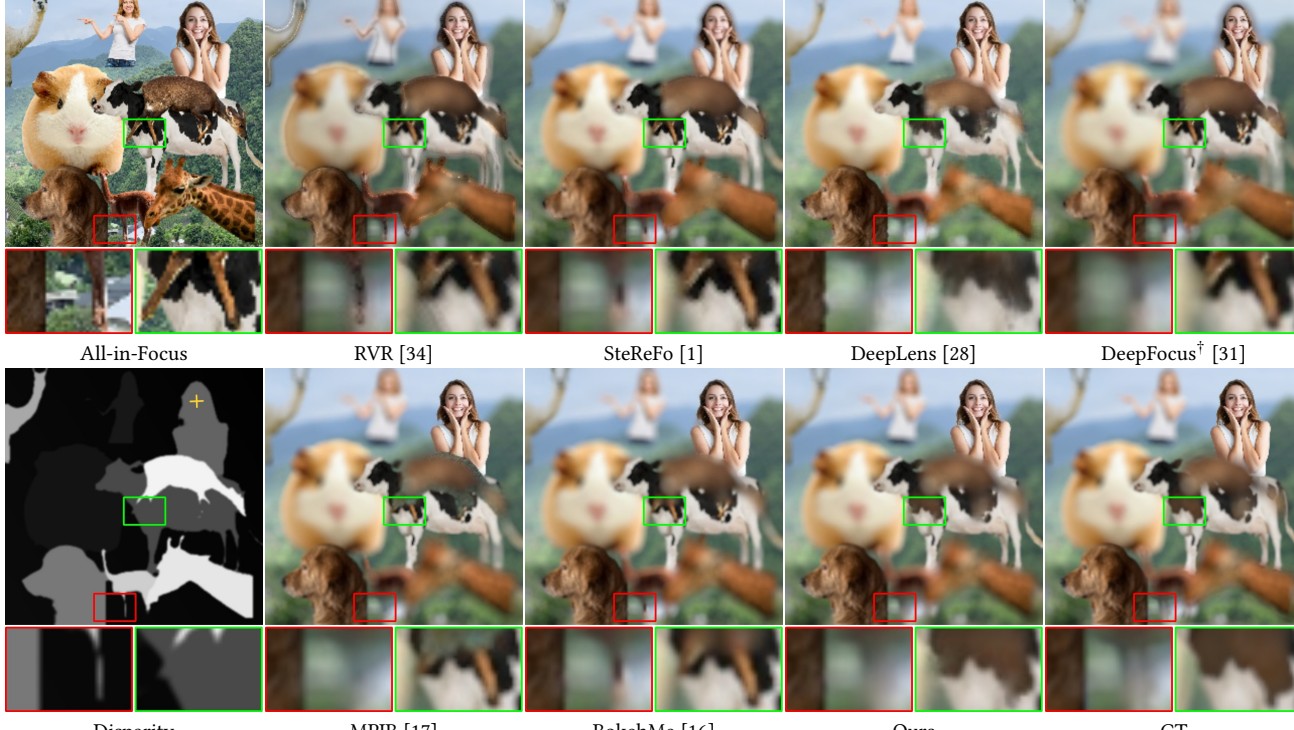

**Figure 6: Qualitative results on the synthetic test set. The focal plane is labeled with a yellow cross on the disparity map.**

**Table 3: Comparison results of the user study.**

| Comparison | Human preference |
| --- | --- |
| Ours vs. RVR [34] | 87.63% / 12.37% |
| Ours vs. SteReFo [1] | 83.16% / 16.84% |
| Ours vs. DeepLens [28] | 71.71% / 28.29% |
| Ours vs. MPIB [17] | 82.32% / 17.68% |
| Ours vs. BokehMe [16] | 64.04% / 35.96% |

each participant is shown 2 videos at a time, comprising two bokeh-rendered videos produced by our approach and a method randomly chosen from RVR [34], SteReFo [1], DeepLens [28], MPIB [17], BokehMe [16], in random order. Participants are asked to choose the method with more consistent and aesthetic bokeh effects or choose none if it was difficult to judge. Additionally, each participant needs to complete at least 10 tests before submitting results. The user study involves 55 participants, and the results in Table. 3 indicate that our method is consistently preferred by most users.

**Qualitative results.** To intuitively show why our method is more frequently preferred by the users, we showcase some examples in Fig. 7. In the first row, the focal plane is on the woman. Other methods mistakenly blur the edges of the hand and lipstick as they are considered as the background in the disparity map(The disparity maps are shown in supplementary materials). Instead, the generated video of VBR perfectly preserves the details on the edges of the hand and the lipstick, demonstrating its robustness against

inaccurate disparity maps. In the second row, the focal plane is on the boy in the mirror. All the other methods produce abnormal artifacts at edges, especially on the boy's hair. Our model generates correct blur results while keeping the edge of the focused object clear. The single-image-based methods can not produce satisfying results as they either overlook the disocclusion phenomenon or utilize an inpainting network (MPIB [17]) to address it in an unstable manner. The results further verify that leveraging information from adjacent frames benefits our model in mitigating the disocclusion phenomenon. See supplementary materials for more visualization and video results.

## 6.4 Ablation Studies

In this section, we explore the effectiveness of each component in the VBR framework, from modules tailored to temporal consistency to augmentation strategies for robustness.

**Effects of Temporal Designs.** Since the main difference between the image-based and the video-based rendering methods lies in their capability to leverage the information from neighboring frames, we explore whether our design can truly benefit video-based rendering. we implement a plain model with only spatial loss (row 1). Under this setting, our model degrades to a single-image-based model, neglecting information from adjacent frames.

As illustrated in Table. 4, through comparing row 1 with row 3 and row 2 with row 5, utilizing the **temporal fusion block** enables the model to achieve superior render quality and temporal consistency. This result demonstrates the effectiveness of the temporal

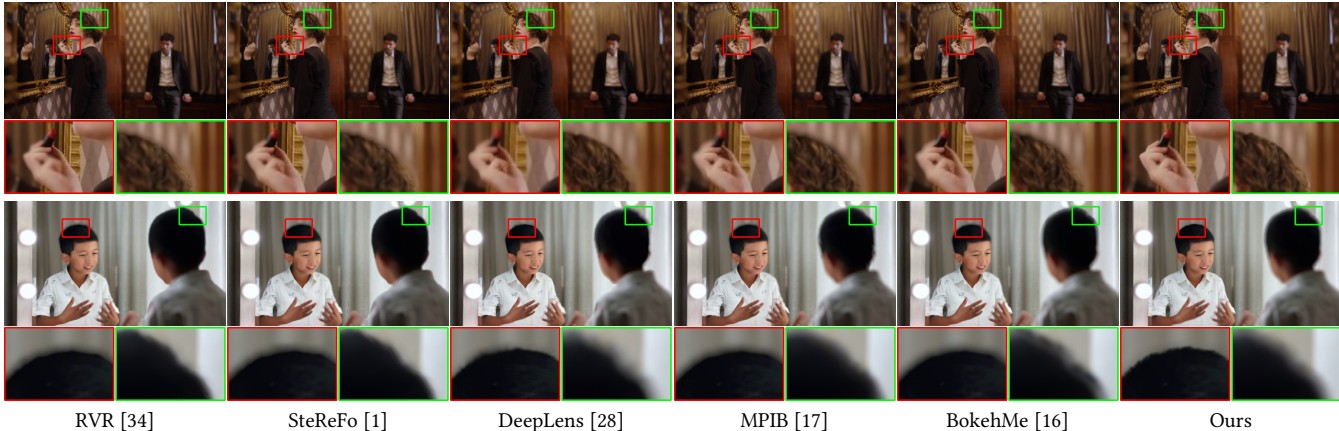

| RVR [34] | SteReFo [1] | DeepLens [28] | MPIB [17] | BokehMe [16] | Ours |

Figure 7: Qualitative results on real-world videos. Please zoom in to see more details.

Table 4: Ablation studies on temporal modeling.

| Settings | PSNR↑ | SSIM↑ | $PSNR_{ob}$ ↑ | $SSIM_{ob}$ ↑ | Consistency↓ |
|---|---|---|---|---|---|
| w/o TFB & $L_r$ | 34.8 | 0.975 | 29.9 | 0.946 | 0.069 |
| w/o TFB | 34.9 | 0.975 | 30.1 | 0.947 | 0.066 |
| w/o $L_r$ | 35.4 | 0.977 | 30.5 | 0.950 | 0.059 |
| w/o DCN | 34.7 | 0.973 | 29.7 | 0.943 | 0.072 |
| Full model | **35.6** | **0.978** | **30.6** | **0.951** | **0.056** |

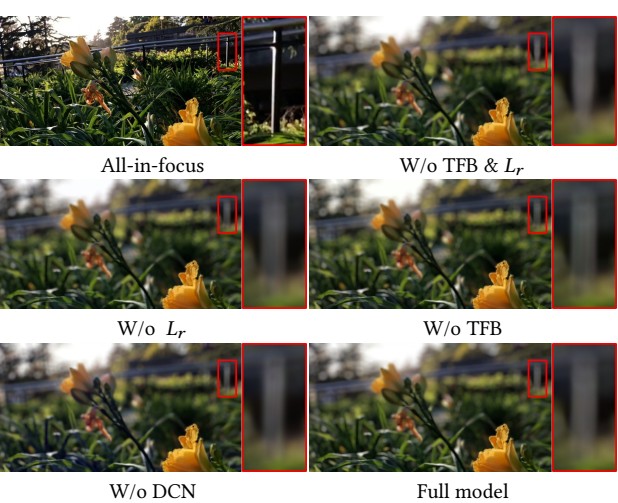

Figure 8: Quantitative result of the ablation study.

fusion block in mitigating flicker between frames and addressing disocclusion issues. Furthermore, we conduct an ablation study on the **deformable convolution** (row 4 vs. row 5) within the temporal fusion block, replacing it with standard convolution to assess the necessity of alignment. The results show that unaligned features pose challenges for integrating information from adjacent frames, highlighting the importance of feature alignment.

To evaluate the effectiveness of **temporal loss**, from Table. 4, one can see that incorporating temporal consistency loss enhances performance across all metrics (row 3 vs. row 5) and can reach a synergy with the temporal fusion block. To intuitively explore the significance of each component, we provide visual examples in Fig. 8. As shown in the figure, the full model yields superior rendering outcomes, especially in edge regions (zoomed-in areas), highlighting the necessity of using the temporal fusion block to leverage temporal information. Furthermore, by constraining temporal consistency with the temporal loss, the output exhibits more evenly distributed blur circles.

**Effects of disparity augmentation.** As depicted in Fig. 5, disparity augmentation strategy plays a crucial role in enhancing the model's robustness to inaccurate disparity maps. Please refer to the supplementary materials for a more in-depth ablation study on disparity augmentation.

## 7 CONCLUSION

In this work, we propose a novel framework named Video Bokeh Renderer (VBR) to generate aesthetic bokeh effects from all-in-focus videos. The framework consists of two sub-modules: a coarse bokeh generator and an iterative bokeh refiner. To mitigate rendering flicker between frames and address the disocclusion phenomenon, we introduce a temporal fusion block into our sub-modules to align and fuse features from adjacent frames. Experimental results show that our method produces rendering results with better temporal consistency, enhanced edge rendering quality, and increased robustness to low-quality disparity maps. This work also contributes a video bokeh rendering dataset, SVB, to alleviate data shortages. To the best of our knowledge, VBR is the first framework to consider temporal information in video bokeh rendering. We hope our work can serve as a solid baseline and inspire further research.

**Limitations and future work.** Although the synthetic bokeh rendering dataset captures authentic bokeh effects, which can be utilized for training video bokeh models, it still exhibits a gap compared to real-world datasets. In our future work, we intend to explore additional methods to generate more realistic datasets.

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
