# OpenReview forum: "Video Bokeh Rendering: Make Casual Videography Cinematic"
_acmmm.org/ACMMM/2024/Conference — MM2024 Oral_

### Official Review · Reviewer_36VE · 2024-05-15

**Rating:** 4
**Confidence:** 3

**Summary:**

The authors propose a novel method to generate bokeh effects from all-in-focus videos (+ disparity maps) , consisting of a coarse bokeh generator and an iterative refiner module. A novel temporal fusion block is added for better temporal consistency (mitigating flicker effects) and handling of disocclusions.

**Strengths:**

- Proposed approach is sound and carries sufficient novelty, especially the temporal fusion component is interesting.
- Experiments and evaluation section is very comprehensive, consisting both of quantitative and qualitative (human preference) comparison studies.
- Evaluation shows that the proposed approach gives better performance than the state of the art, both quantitatively and qualitatively.

**Limitations:**

- The topic of bokeh effect rendering is quite a niche topic in the multimedia field.
- The synthetically generated video dataset (SVB dataset) has a very small resolution of only 256x256 pixels.
   This is not representative of modern video content, which is at least Full HD.
   Therefore, it is not clear whether the quantitative results from table 1 (utilizing the SVB dataset) apply also for modern video content.

**Suitability:**

2

---

### Official Review · Reviewer_Jct2 · 2024-05-18

**Rating:** 6
**Confidence:** 4

**Summary:**

This paper focuses on video bokeh rendering and constructs a novel datasets. The theoretical modeling and network are detailed, and sufficient experiments prove the superior performance of the VBR.

**Strengths:**

1. The experiments are sufficient and convincing.
2. Writing, figures and tables are satisfying.
3. The first video bokeh rendering dataset is constructed, and the proposed solution has excellent performance.

**Limitations:**

1. How does the method generate more obvious bokeh effects outside the training set. Whether the actual maximum blur effect is limited by the training set?
2. Reporting computational costs can further refine the manuscript.
3. More works in relation to bokeh rendering should be discussed. [1] Zheng B, Chen Q, Yuan S, et al. Constrained predictive filters for single image bokeh rendering[J]. IEEE Transactions on Computational Imaging, 2022, 8: 346-357. [2] Mandl D, Mori S, Mohr P, et al. Neural Bokeh: Learning Lens Blur for Computational Videography and Out-of-Focus Mixed Reality[C]//2024 IEEE Conference Virtual Reality and 3D User Interfaces (VR). IEEE, 2024: 870-880.
4. This work is beneficial to the Bokeh community, looking forward to codes and datasets.

**Suitability:**

3

---

### Official Review · Reviewer_nBQj · 2024-05-24

**Rating:** 5
**Confidence:** 3

**Summary:**

This paper introduces the "Video Bokeh Renderer" (VBR), a model that transforms all-in-focus videos into ones with shallow depth of field to simulate cinematic background blur. Unlike traditional methods requiring professional equipment, VBR can achieve similar effects using common smartphones. The model addresses issues like flickering and edge artifacts in video by enhancing temporal consistency and utilizing inter-frame information. Additionally, the paper presents a synthetic video bokeh dataset (SVB) to overcome training data shortages.

**Strengths:**

The "Video Bokeh Renderer" (VBR) is a new approach that extends bokeh rendering from static images to dynamic video content, addressing the challenge of maintaining temporal consistency across frames;
The methodology described is technically sound, with a clear explanation of how the model utilizes both the spatial and temporal information to generate high-quality bokeh effects in videos;
The paper is well-structured and clearly written, making complex concepts accessible and understandable.

**Limitations:**

The reliance on the synthetic dataset (SVB) for training and validation might limit the model’s performance in real-world scenarios where lighting, motion, and other environmental factors are more complex and less predictable than in synthetic datasets;
The paper could explore further how well the model generalizes across different types of videos involving various motion dynamics, backgrounds, and lighting conditions. There may be a need for more extensive testing on diverse datasets to validate the robustness of the model under varied conditions;
The methodology and demographic details of the user study might require more depth to ensure that the preferences and perceptions are representative of the target audience.

**Suitability:**

3

---

### Official Review · Reviewer_feW3 · 2024-05-25

**Rating:** 4
**Confidence:** 3

**Summary:**

The paper proposes a novel framework named Video Bokeh Renderer (VBR) to generate aesthetic bokeh effects from all-in-focus videos. Experimental results show that VBR produces rendering results with better temporal consistency, enhanced edge rendering quality, and increased robustness to low-quality disparity maps. Besides, this paper also contributes a video bokeh rendering dataset, SVB, to alleviate data shortages.

**Strengths:**

It is a topic of interest to researchers in related areas. The paper has a lot of potential contributions: the dataset, the content analysis, and the explorative user study. Moreover, the authors intend to conduct a decent experiment, except for the rationale for selecting the baseline algorithms.

**Limitations:**

Overall, the context part is easy to read; however, several improvements can be made to increase the quality and contribution of the paper. My detailed comments are as follows.
1.	Page 1, Line 115, this paper aims to bridge the gap between smartphones and cameras, making casual videography cinematic. However, the method proposed will introduce additional overhead of smartphones. Conducting performance testing on power consumption will make this work more convincing.
2.	Page 5, Line 540-545, this paper contributes a dataset consisting of 3000 videos. Please briefly describe the categorizations of these videos and include some graphs for demonstration.
3.	This paper selects MPIB and BokehMe as the latest baselines which are proposed in 2022. Is there any related method in 2023 and 2024?
4.	Page 6, Section 6.2, when conducting quantitative analysis of the results, providing the percentage of improvement helps to enhance the understanding of the effectiveness of this method and make it more convincing.
5.	Page 6, Section 6.2, this paper compares the results on the SVB dataset of the proposed method with baselines. However, the analysis of the results is not comprehensive. For example, can the better robustness of this model be explained from the perspective of its model design?

Minor comments
1.	Page 8, Table 4 and Figure 8, please maintain consistent capitalization of the usages of “W/o”. Please explain the meanings of w/o briefly.
2.	The capitalization of subsection titles is inconsistent. For example, "2.1 Bokeh Rendering" is different from "3.1 Disocclusion phenomenon".

**Suitability:**

2

---

### Meta-Review · Area_Chair_9WSS · 2024-07-02

**Recommendation:** Accept (Oral)
**Confidence:** 4

**Metareview:**

The paper is well written and clearly presents the proposed dataset and rendering approaches.
Out of the 4 reviewers, 1 proposes a definite accept, 2 a weak accept, 1 a borderline accept.
Overall, all are very positive.
Only reviewer 4 is more critical, with criticism mainly addressing the aspects of (1) the topic being niche and (2) the resolution addressed in the paper not being practically realistic.
The authors provide a comprehensive rebuttal. Some of the additional analyses, such as the one regarding resolution-stability, should be included in the camera-ready version of the paper.
Apparently not yet including additional information in the final paper version is what slightly reduces my "confidence" in my "accept" vote. In case this information from the rebuttal shall be included, I see very good reasons to accept the paper.
The concern that the paper is "niche" is not shared by me. In my view, as Bokeh and DoF rendering are highly relevant aspects in image and video capture and appeal perception. Also, a conference like ACM MM can benefit from some diversity of topics, which this quite high-quality paper can contribute to.

The positive points given by the reviewers can be summarized as follows:

(feW3)

- topic of interest to researchers in related areas.
- paper has lot of potential contributions: dataset, content analysis,  explorative user study.
- Moreover,  authors intend to conduct a decent experiment, except for rationale for selecting the baseline algorithms.

(nBQj)

- "Video Bokeh Renderer" (VBR) is new approach that extends bokeh rendering from static images to dynamic video content
- addresses  challenge of maintaining temporal consistency across frames
- technically sound
- clear explanation of how the model utilizes both the spatial and temporal information to generate high-quality bokeh effects in videos
- paper is well-structured and clearly written, making complex concepts accessible and understandable.

(Jct2)

- The experiments are sufficient and convincing.
- Writing, figures and tables are satisfying.
- The first video bokeh rendering dataset is constructed, and the proposed solution has excellent performance.

(36VE)

- Proposed approach is sound and carries sufficient novelty, especially the temporal fusion component is interesting.
- Experiments and evaluation section is very comprehensive, consisting both of quantitative and qualitative (human preference) comparison studies.
- Evaluation shows that the proposed approach gives better performance than the state of the art, both quantitatively and qualitatively.

For the negative points, most have been addressed in the rebuttal.
As stated above, it is somewhat unclear to which extent the explanations in the rebuttal will be implemented in the camera-ready version of the paper.
At this stage, this is the only concern I have with the paper.

Hence, in case of final acceptance of the paper, authors are strongly encouraged to make sure that all points mentioned in the rebuttal and further reviewer comments are considered for the camera-ready version of the paper.